# Material Evaluation and Dynamic Powder Deposition Modeling of PEEK/CF Composite for Laser Powder Bed Fusion Process

**DOI:** 10.3390/polym15132863

**Published:** 2023-06-28

**Authors:** Jiang Li, Fulun Peng, Hongguang Li, Zhibing Ru, Junjie Fu, Wen Zhu

**Affiliations:** 1Xi’an Institute of Applied Optics, Reconnaissance Vehicle R&D Center, Xi’an 710065, China; redlight007@163.com (H.L.); ruzhb@126.com (Z.R.); fujunjie@stu.xjtu.edu.cn (J.F.); 2State IJR Center of Aerospace Design and Additive Manufacturing, Northwestern Polytechnical University, Xi’an 710072, China; wenzhu@mail.nwpu.edu.cn

**Keywords:** laser powder bed fusion, composite polymers, discrete element method, material characterization, additive manufacturing

## Abstract

Polymeric composites such as Poly-ether-ether-ketone (PEEK)/carbon fiber (CF) have been widely utilized due to outstanding performances such as high specific strength and specific modulus. The PEEK/CF components via powder bed fusion additive manufacturing usually show brittle fracture behaviors induced by their poor interfacial affinity and inner voids. These defects are strongly associated with powder packing quality upon deposition. The particle dynamic model has been widely employed to study the interactions of particle motions. Powder property, bulk material property, and interfacial features of composite powders are key factors in the particle dynamic model. In this work, an efficient and systematic material evaluation is developed for composite powders to investigate their deposition mechanism. The discrete element method is utilized to simulate the dynamic behaviors of PEEK/CF composite powders. The powder properties, bulk material properties, and interfacial features of powders are calibrated and justified by experimental measurement, numerical simulation, and design of experiments. The particle dynamic model can explain the powder flow behaviors and interactions. The experimental and simulation AOR results show a maximal deviation of 4.89%. It reveals that the addition of short CF particles can assist the flow of PEEK powders and improve the packing quality of the composite powders. The results show an experimental improvement of 31.3% and 55.2% for PEEK/CF_30wt% and PEEK/CF_50wt%, with a simulated improvement of 27.4% and 50.2% for corresponding composite powders.

## 1. Introduction

Laser powder bed fusion (LPBF) technology, as an essential category of multiple additive manufacturing technologies, is able to produce polymeric composite materials economically and with obvious flexibility [1,2]. This technique is widely used in aerospace [3,4], automotive manufacturing [5,6], optical components [7,8,9], and other industrial fields. The technique of LPBF uses a high-energy intensity laser beam to selectively melt the powder, which can fabricate more complex and superior structures due to its smaller laser spot and thinner layer thickness [10,11,12]. Polymeric composites such as Poly-ether-ether-ketone (PEEK)/carbon fiber (CF, specific chopped fiber in this work) have attracted the attention of researchers as well as industrialists due to outstanding performances such as high specific strength and specific modulus. However, the laser-sintered PEEK/CF components are usually brittle because of their poor interfacial affinity and inner voids. These defects are strongly associated with the powder packing quality upon deposition. Thus, the investigation of the powder deposition process incorporating the detailed material properties is necessary, so that the expected behaviors of the laser-sintered PEEK/CF composites can be achieved.

The particle dynamic model has been widely employed to study the interactions between particles in macro-scale or micro-scale, especially the powder deposition process [13,14]. The powder property, bulk material property, and interfacial features of composite powders are key factors influencing the packing quality of deposited layers in the LPBF process. Thus, the detailed material evaluation for the dynamic model is significant for revealing the deposition mechanism of composite powders. The evaluation of specific materials has been carried out, such as agriculture [15,16,17], soil [18,19], and rock [20,21,22]. However, little research has conducted the raw material evaluation correlating with the polymer and polymer-based composite powders. Unlike pure materials, the multi-component composite powders will improve the quantities of to-be-determined constants. It also brings some parameters that cannot be measured directly, such as surface energy.

When exploring these specific properties, the reliable approach is to perform either in situ measurements or laboratory experiments. Vu-Quoc et al. [23] measured the restitution coefficient in soybeans by conducting drop tests from various heights. They used the data from high-speed video to determine the final parameters. Marczewska et al. [24] utilized the triaxial test to investigate the effects of particle-particle sliding friction and contact stiffness, including Young’s modulus, Poisson’s ratio, and dilatancy. Other experimental measurements such as uniaxial tension, slope test, and direct shear test also play an important role. However, it is difficult to measure the specific contact parameters in CF-reinforced composite powders. Preparing the pure CF sample is a struggle, causing inconvenience in the corresponding experiments. Thus, it is appropriate to compute the specific coefficient by the finite element simulation in macro-scale. Some parameters cannot be measured by experiments such as surface energy, which can be determined by the design of experiments (DOE) method. Yoon et al. [20] employed the response surface method and the Plackett–Burman (PB) design to acquire suitable parameters to simulate the uniaxial compression of bonded rock. Johnstone [25] used the DOE method to calibrate the dynamics model based on experimental measurements. E. X. Zu et al. [26] carried out the simulation of coke accumulation based on the Hertz–Mindlin (no-slip) contact model. The core of their study is to determine the contact parameters. The number of investigated factors has a great influence on the complexity of the DOE. Therefore, the number of studied factors should be reduced as much as possible, especially for the composite powders. Moreover, the multi-phase characteristics of polymer composite powders significantly increase the number of undetermined parameters, causing difficulties in calibrating the DEM parameters. Thus, this work proposes a parameter design method called the “Plackett–Burman (PB) Box–Behnken design (BBD) Genetic Algorithm (GA)” for the DEM model parameter calibration process. This method mainly aims at contact parameters that are difficult to measure experimentally. It can quickly obtain the functional relationship between the significant parameters and the target value and determine the optimal parameter combination. Importantly, this method possesses universality for multi-phase powder systems.

In this work, to develop a reliable powder dynamics model for powder deposition in LPBF, a systematic framework for material evaluation of PEEK/CF powders with different compositions is proposed. Therein, the powder properties, bulk material properties, and interfacial features are calibrated via experimental measurement, numerical simulation, and the DOE method. The powder dynamics model for PEEK/CF powders is experimentally verified by the fluidity-reflected angle of repose (AOR). Ultimately, the powder deposition processes of PEEK and PEEK/CF powders are evaluated and analyzed through the experiments and the validated dynamics model.

## 2. Discrete Element Method

### 2.1. Contact Model Theory

The discrete element method (DEM) is utilized to simulate the dynamic behaviors of PEEK/CF composite powders. In the DEM simulation, the particles are modeled as discrete entities to describe the dynamics of particle flow. The translation, rotation, and location of the particles can be characterized by Newton’s laws of motion, which is described as follows [27,28]:(1)$$m_{i}\frac{dv_{i}}{dt}=\sum F_{c,i}+m_{i}g$$
(2)$$\frac{d\left(I_{i}\cdot \omega_{i}\right)}{dt}=R_{i}\cdot \sum M_{c,i}$$
where *m_i_*, ***ω****_i_*, ***v****_i_*, and ***I****_i_* are the mass, angular velocity, translational velocity, and moment of inertia, respectively. ***F****_c,i_* is the contact force, ***M****_c,i_* is the contact torque, and ***R****_i_* is the rotation matrix.

The relationship between cohesive force and particle size is shown in Figure 1a [29,30]. It appears that the cohesive forces such as Van der Waals force, electrostatic force, and capillary force are greater than the gravity of particles when the particle size is less than 100 μm, which reduces the powder fluidity and causes inconvenience for the powder-spreading process. In reality, the powder is preheated and dried before the powder-spreading process. Thus, the capillary force can be ignored.

(1)Hertz–Mindlin contact model

The rationality of the Hertz–Mindlin contact model has been widely recognized in the particle-related research field [31,32]. In order to accurately characterize the stress–strain relationship between particles considering adhesion, this work adopts the Hertz–Mindlin and Johnson–Kendall–Roberts (JKR) model for polymers and their composite powders, which is implemented using EDEM software. Among them, the Hertz–Mindlin model is the foundation of the DEM method, and the parameter of surface energy in the JKR model can comprehensively characterize the bonding effect between particles caused by the van der Waals force and the electrostatic force [33]. Therefore, the Hertz–Mindlin and JKR models possess universality in the DEM simulation of other thermoplastic polymers and their composite powders.

In the Hertz–Mindlin model, the normal and tangential force components between particles are calculated based on the Hertzian contact theory and the Middlin–Deresiewicz method, respectively [34,35]. Both normal and tangential force between particles have damping components. The damping coefficient is related to the restitution coefficient [36]. The tangential friction force between particles follows Coulomb’s law of friction, and the rolling friction between particles is computed based on the Contact-Directional-Constant-Torque model [27,37].

Figure 2 shows the schematic diagram of the Hertz–Mindlin model for adjacent particles *i* and *j*, where the normal contact force on particle *i* can be expressed as
(3)$$F_{CN}=F_{CN,E}+F_{CN,D}$$
where ***F****_CN,E_* is the normal elastic force, and ***F****_CN,D_* is the normal damping force. The ***F****_CN,E_* is computed based on the Hertz contact theory, which is described as
(4)$$F_{CN,E}=-K_{n}\times \alpha_{n}^{1.5}$$
where ***α****_n_* is the normal contact displacement (overlap of adjacent particles). The Contact constant *K_n_* is expressed as
(5)$$K_{n}=\frac{4G\sqrt{r}}{3\left(1-\upsilon \right)}$$
where *G* is the shear modulus of the particle, *r* is the radius of the local curvature, and *υ* is the Poisson’s ratio.

The normal damping force ***F****_CN,D_* between particles is expressed as
(6)$$F_{CN,D}=-2\sqrt{\frac{5}{6}}b^{*}\sqrt{m^{*}\times k_{n}}\times v_{re,n}$$
where ***v****_re,n_* is the component of the relative velocity of particle *i* relative to particle j along the normal direction. *k_n_* is the normal contact stiffness, *β^*^* is the damping coefficient, and *m^*^* is the equivalent mass. Therein, the damping coefficient and normal contact stiffness can be expressed as
(7)$$\beta^{*}=\frac{lne}{\sqrt{ln^{2}e+\pi^{2}}}$$
(8)$$k_{n}=\frac{2G}{1-\upsilon }{\left(\frac{3\left(1-\upsilon \right)\times r\times F_{CN,E}}{4G}\right)}^{\frac{1}{3}}$$
where *e* is the restitution coefficient between particles. The equivalent mass *m^*^* and equivalent curvature radius *r* of particles are, respectively, described as
(9)$$m^{*}=\frac{m_{i}\times m_{j}}{m_{i}+m_{j}}$$
(10)$$r=\frac{r_{i}\times r_{j}}{r_{i}+r_{j}}$$
where *m_i_* and *m_j_* represent the masses of adjacent particles *i* and *j*. *r_i_* and *r_j_* are the radiuses of adjacent particles *i* and *j*.

The tangential contact force ***F****_CT_* between adjacent particles can be expressed as
(11)$$F_{CT}=F_{CT,E}+F_{CT,D}$$
where ***F****_CT,E_* is the elastic force along the tangential direction, and ***F****_CT,D_* is the damping force along the tangential direction. The tangential elastic force ***F****_CT,E_* can be expressed in incremental form as
(12)$$F_{CT,E}=F_{CT,E,\left(n-1\right)}+\Delta F_{CT,E}$$
where ***F****_CT,E,_*_(*n*−1)_ is the tangential elastic force of the previous analysis step; ∆***F****_CT,E_* is the increment of the tangential elastic force in the current analysis step, which is described as
(13)$$\Delta F_{CT,E}=k_{t}\times \Delta \alpha_{t}$$
where ∆***α**_t_* is the tangential displacement increment, and *k_t_* is the tangential spring stiffness, expressed as
(14)$$\Delta \alpha_{t}=v_{re,t}\times \Delta t$$
(15)$$k_{t}=\frac{2}{2-\upsilon }{\left(6\left(1-\upsilon \right)\times G^{2}\times r\times F_{CN,E}\right)}^{\frac{1}{3}}$$
where ***v****_re,t_* is the relative tangential velocity between particles, and ∆*t* is the step time. Meanwhile, the tangential damping force between adjacent particles can be expressed as
(16)$$\Delta F_{CT,D}=-2\sqrt{\frac{5}{6}}\times b^{*}\times \sqrt{m^{*}\times k_{t}}\times v_{re,t}$$

It is worth noting that the tangential force between adjacent particles is related to the frictional force between particles (*μ_s_**F**_CN_*, *μ_s_* is the static friction coefficient). In the simulation process, the rolling friction cannot be ignored, which is achieved by applying a torque to the contact surface of the particles:(17)$$\tau_{i}=-\mu_{r}F_{\mathrm{CN}}\times R_{i}\times \omega_{i}$$
where *R_i_* is the distance from the contact point to the mass center, *μ_r_* is the rolling friction coefficient, and ***ω****_i_* is the unit angular velocity vector at the contact point.

(2)JKR theory

The Hertz–Mindlin with JKR contact model can form an adhesive force contact model between particles, mainly considering the influence of van der Waals forces in the contact area. In this model, the normal elastic contact force between particles is computed based on the JKR theory [38], including the overlap of particles δ, interaction parameters, and surface energy *γ*, which can be described as
(18)$$F_{JKR}=-4\sqrt{\pi \gamma E^{*}}\alpha^{\frac{3}{2}}+\;\frac{4E^{*}}{3r}\alpha^{3}$$
(19)$$\delta =\frac{\alpha^{2}}{r}-\sqrt{\frac{4\pi \gamma \alpha }{E^{*}}}$$
(20)$$\frac{1}{E*}=\frac{\left(1-\upsilon_{i}^{2}\right)}{E_{i}}+\frac{\left(1-\upsilon_{j}^{2}\right)}{E_{j}}$$
where *E*^*^ is the equivalent Young’s modulus, *r*^*^ is the equivalent curvature radius, and *α* is the contact radius of particles.

It is discovered that several parameters are involved in the aforementioned model, such as static friction coefficient, rolling friction coefficient, restitution coefficient, and surface energy. Therefore, it is essential to carry out the parameter calibration work through the material characterization.

### 2.2. Parameter Classification of the Discrete Element Model

The first step of the DEM simulation is to calibrate numerous parameters through comprehensive material characterization. Figure 3 shows the parameters involved in the DEM model, which can be divided into three categories:(1)Solid-phase parameters: solid phase density, elastic modulus, and Poisson’s ratio;(2)Contact parameters between particles: static friction coefficient, rolling friction coefficient, restitution coefficient, and surface energy;(3)Powder-phase parameters: particle morphology, particle size, and powder fluidity.

For composite powders, the contact forms between particles can be further divided into three patterns of “particle-particle”, “particle-reinforced phase”, and “reinforced phase-reinforced phase”. Specifically, the particle size and the static friction coefficient between particles are directly measured. The restitution coefficient is obtained via the FEM simulation. The rolling friction coefficient and surface energy are achieved using the “PB-BBD-GA” parameter-design method. This method extends the GA on the basis of traditional PB and BBD methods, avoiding the uncertainty of traditional methods in predicting the optimal parameter combination. The “PB-BBD-GA” method can quickly obtain the functional relationship between significant parameters and target values and determine the unknown parameter combination more accurately and efficiently.

### 2.3. Analysis Step and Particle Modeling

The DEM simulation is proceeding based on the time step. In the numerical computation, a differential equation needs to be transformed into a difference equation for the iterative solution. The time step is the time difference between two adjacent iterations. The time step is usually set to 20–30% of Rayleigh time, which makes the simulation stable and timesaving. The Rayleigh time is determined as [39]
(21)$$T_{R}=1.625R_{min}\sqrt{\frac{\rho }{E}}$$
where *R_min_* is the minimum radius of spherical members and *ρ* is the density. In this work, the Rayleigh time is 6.4175 × 10^−9^ s and the DEM results are recorded every 0.001 s.

The particle model is established based on its morphology in the DEM simulation, and the spherical particles are computationally efficient compared with the non-spherical particles. Importantly, the contact model for spherical particles should include rolling resistance to model the rotational motion of actual non-spherical particles. In contrast, the rolling resistance can be omitted when utilizing the non-spherical particles [40]. In this work, the spherical particles are modeled and the rolling resistance represented by the rolling friction coefficient is determined last. Meanwhile, a multi-sphere method, by which multiple spheres with the same or different sizes are clumped to approximate the irregularly shaped particle (such as CF in this work), is employed in the model [41], as shown in Figure 1c.

## 3. Material and the Experimental Method

### 3.1. Material Characterization

This work selects the materials of PEEK (ZYPEEK_330PF, Jilin Joinature Polymer Co., Ltd., Changchun, China) and PEEK/CF (CF from NanJing WeiDa Composite Material Co. Ltd., Nanjing, China) powders, which possess the features of poor powder fluidity and high difficulty in powder deposition process, as the representative research object. The adopted powders are the PEEK, PEEK/CF_30wt%, and PEEK/CF_50wt%, which are featured by the weight fraction of CF. The inherent properties of the compositions of PEEK and CF are listed in Table 1, which includes important input factors of the subsequently introduced powder dynamics model.

As shown in Figure 3, a large number of constants need to be determined, which are described in detail as follows.

(1)Angle of repose

Powder fluidity is a comprehensive characterization that deeply impacts the powder deposition process. The AOR can reflect the powder fluidity in a simple and convenient fashion. In this paper, the AOR is measured according to GB11986-89. As shown in Figure 4, a funnel is fixed above the horizontal plate, through which the particle sample falls, and the AOR of the cone is measured. Both experiments and the DEM model follow this standard.

(2)Particle morphology and particle size distribution

Particle morphology and particle size distribution have important impacts on powder fluidity and computational efficiency. The scanning electron microscope (SEM, TM4000PLUS, HITACHI, Tokyo, Japan) equipment is utilized in the high vacuum circumstance and at a voltage of 5 kV to acquire the SEM images that characterized the morphological features of PEEK powders and PEEK/CF composite powders. The objective area of the SEM image is limited. Therefore, the Malvern laser particle sizer (Mastersizer 2000, the Malvern Panalytical Ltd, Marvin City, UK) is utilized to measure the accurate particle size distribution of PEEK and CF.

(3)Bulk density

The measurement method for the bulk density of PEEK and PEEK/CF powders is shown in Figure 5, which refers to GB/T 16913.3-1997. Prepare a 100 mL graduated cylinder and place the funnel at a certain height directly above the cylinder. The powder falls freely from the funnel mouth and fills the cylinder. Thereafter, screed the surface of the graduated cylinder, place it on the electronic scale to measure its mass, and obtain the bulk density of the powders.

(4)Static friction coefficient

The inclined plane test can be used to measure the static friction coefficient. The friction coefficient between different materials can be achieved by replacing the material of the inclined plane or sample. At first, the inclined plane is placed horizontally, and the sample is placed at the fixed position of the inclined plane. Then, lift the slope slowly, and stop the inclined plane when the sample starts to slide. The position of the inclined plane at this moment is recorded through a camera, and the tilt angle is read. The static friction coefficient is the tangent value of the inclination angle. It is worth noting that when measuring the static friction coefficient between powders, it is necessary to evenly sprinkle some corresponding powder samples on the inclined surface of the same material, which makes the results more accurate.

(5)Restitution coefficient

The restitution coefficient is the ratio of the separation velocity and the approaching velocity of the two objects along the normal direction of the contact before and after the collision, which is only related to the material of the collision objects [44]. Generally, it can be experimentally measured. Fix the plate, and a ball falls freely above the plate. The coefficient of restitution can be calculated by recording the fall height and bounce height of the ball (the restitution coefficient between different materials is achieved by changing the materials of the plate and the free-falling ball). However, measuring the restitution coefficient associated with pure CF is a struggle due to the difficulty in producing the products in pure CF in reality. Thus, it is appropriate to calculate the restitution coefficient using the simulation. The process of free-falling and bouncing for a small ball is achieved via FEM simulation (as shown in Figure 6) [45]. The restitution coefficient is computed as
(22)$$e=\frac{v_{off}}{v_{app}}$$
where *v_app_* is the speed of the ball approaching the plate, *v_off_* is the speed of the ball away from the plate, and *e* denotes the restitution coefficient.

(6)Rolling friction coefficient and surface energy

Experimentally measuring the rolling friction coefficient and surface energy is a struggle and can even be impossible to achieve. Thus, the DOE including the PB, BBD, and GA is employed to determine them.

PB design is a two-level (−1 and +1) experimental design method, which attempts to screen and determine the factors that significantly impact the results with the least number of trials. In this work, AOR is used as the response value to screen eight factors corresponding to the powders of PEEK/CF_30wt%. These factors with their initial parameter spaces are listed in Table 2. When determining the parameter space, the ratio of high and low levels of the two parameters is set to be identical in order to prevent significant fluctuation between the investigated factors [46]. The detailed design and results of the PB design are discussed in Section 4.2.

The BBD design is utilized to further investigate the optimal points of the key factors based on the PB results. The computed results are analyzed afterward by the response surface method. The corresponding variables are coded as follows:(23)$$X_{i}=\left(x_{i}-x_{0}\right)/{\Delta x}_{i}\left(i=1,2,...,k\right)$$
where *X_i_* is the coded value of the independent factor, *x*_0_ is the real value at the center point, *x_i_* is the real value, and ∆*x_i_* is the value of step-change.

The BBD model can be described by the quadratic equation:(24)$${Y=\beta }_{0}+\sum \beta_{i}X_{i}+\sum \beta_{ii}X_{i}^{2}+\sum \beta_{ij}X_{i}X_{j}\left(i=1,2,...,k\right)$$
where *Y* is the predicted response value (AOR in this work), *X_i_* and *X_j_* are the coded independent variables, *β_0_* is the offset term, *β_i_* is the linear term, *β_ii_* is the squared term, and *β_ij_* is the interaction term.

GA is developed by imitating the mechanism of natural biological evolution. It is an efficient, parallel, global searching method, which can automatically acquire and accumulate knowledge about the searching space, and the searching process is adaptively controlled to obtain the optimal solution. The entire GA is achieved based on the Matlab GA toolbox in this work.

### 3.2. Powder Deposition Process

The reliability of the complete model is supposed to be experimentally verified. The comprehensive indicator AOR can be used as the target, which reflects the overall performance of the studied powder. In addition, this work investigates the effect of CF on the surface properties of the deposited powder layer through powder deposition experiments (similar to the scraper form of the actual LPBF process) and corresponding simulation.

The surface roughness is the key factor in the powder deposition process. It has a great influence on the sintering process and the subsequent recoating process. Thus, it is necessary to investigate powder deposition through the powder dynamics model. By comparing with the corresponding experiment, the powder dynamics model can be further verified by this practical application.

The powder deposition is experimentally conducted based on a piece of slotted glass, which is shown in Figure 7a. The preparation steps are as follows: (1) prepare a piece of glass with a groove of 0.8–1 mm; (2) load enough powder manually in front of the groove; (3) use another piece of glass to scrape the loaded powders slowly from left to right with a tilt angle of around 45°; (4) the surface of the powders in the groove is smooth to the naked eye, as shown in Figure 7b; (5) the surface morphology of the target zones (Figure 7c) is measured using a microscope (J6000, Keyence, Shanghai, China). The observed area is located at the center of the powder layer.

In the DEM model, set the same conditions as the experiment. When the simulation of the powder deposition process is completed, obtain the surface particle contour curve at the center of the powder layer (as shown in Figure 7d). The fluctuation of the contour curve reflects the flatness of the powder surface.

## 4. Results and Discussion

This section firstly describes the material properties of PEEK and PEEK/CF in detail, including particle morphology, particle size, and AOR, determines the contact parameters between particles (such as rolling friction coefficient, restitution coefficient, and surface energy), and experimentally verifies the reliability of the determined parameters. Secondly, the powder deposition process is investigated through the experiment and the DEM simulation. The influence of CF particles on the surface performance of the powder deposition layer is analyzed.

### 4.1. Material Evaluation

The particle size distribution is obtained as input information in DEM, and it significantly influences computational efficiency. The SEM pictures of pure PEEK and PEEK/CF_30wt% are given in Figure 8. It shows that the powder particles of PEEK are almost ellipsoidal. The diameter of CF is uniform, but the length distribution is obviously dispersed. To specifically characterize the particle size information, the powders of PEEK and CF are detected through the laser particle sizer, and the results are given in Table 3 and Figure 9.

It shows that the average particle size of PEEK is larger than that of CF. The majority of PEEK particles are in the range of 30–60 μm, while the lengths of numerous CF particles are scoped by 5–20 μm.

The restitution coefficient and the static friction coefficient are directly acquired according to the finite element simulation and inclined plane test, respectively. The corresponding results are summarized in Table 4. It appears that the PEEK-related static friction coefficients reach higher values compared with the CF-CF, which is caused by the irregular shapes and unsmooth surface property of the PEEK particles. The glass-related static friction coefficients are minimal due to the glossy surface. As for the restitution coefficient, the PEEK particles possess good elasticity, inducing the PEEK-related restitution coefficients to present larger values than the CF-related restitution coefficients.

The bulk density results of PEEK and PEEK/CF powders are shown in Table 5. It is observed that the bulk density of PEEK/CF is higher than that of the PEEK powder, and the bulk density of PEEK/CF increases with the increase in the CF mass fraction. Because the solid-phase density of CF is slightly higher than that of the PEEK, most CF particles have smaller sizes, filling the pores between PEEK particles.

### 4.2. Results of DOE

The rolling friction coefficient and surface energy cannot be directly measured through experimental methods, so the “PB-BBD-GA” method is used to acquire the optimal combination of rolling friction coefficient and surface energy. Therein, the PB design aims to determine the significant parameters of the related rolling friction coefficient and surface energy. The detailed design and response results of the PB test are listed in Table 6. The significance analysis of the model and corresponding factors is given in Table 7.

As shown in Table 7, both R2 and Adjusted-R2 are close to 1, which means that the model and the chosen response value of AOR are reasonable. The significance can be reflected by the *p*-value. The *p*-value less than 0.05 correlates with the significant factor. Therefore, the parameters of PEEK-PEEK rolling friction coefficient (*X*_0_), PEEK-CF rolling friction coefficient (*X*_1_), and PEEK-PEEK surface energy (*X*_5_) have significant effects on the AOR. Among them, the influence of *X*_0_, *X*_1_, and *X*_5_ are in descending order, which can be obtained through the F-value. This indicates that the effect of the rolling friction coefficient is greater than that of surface energy in this work. Since the ellipsoidal particles of PEEK are replaced by round particles, the rolling friction coefficient should be utilized to make up for the shape approximation in the powder dynamics simulation. Analogous treatment is also found in the literature of [47,48]. Regarding the parameter of rolling friction coefficient/surface energy, the general influence order of PEEK-PEEK, PEEK-CF, and CF-CF is in descending trend, which means that the interaction between CF particles is minimally effective in the composite powders PEEK/CF.

Since the significant parameters (*X*_0_, *X*_1_, *X*_5_) are determined by the PB test, the remaining insignificant parameters can be reasonably fixed before the BBD design. The parameters of *X*_2_, *X*_3_, and *X*_4_ are fixed as 0.01, *X*_6_ and *X*_7_ are set to 0.001. Sequentially, the detailed BBD design (correlated with the parameters of *X*_0_, *X*_1_, and *X*_5_) and response-value of AOR are listed in Table 8. The variance analysis of the whole model and the corresponding factors are given in the following Table 9. Herein, the insignificant quadratic and cross-terms are eliminated for the convenience of analysis and subsequent calculation. The *p*-value of the model is extremely small, which shows that the entire model is reliable and the AOR can be reasonably predicted. Interestingly, the significance of cross-terms is greater than that of the quadratic term (eliminated here) in this numerical example.

Based on the results of the BBD design, the quadratic regression model for the response value (AOR) and the investigated variables (*X*_0_, *X*_1_, *X*_5_) can be established as:(25)$$\begin{array}{l} {\mathrm{ROE}}_{\mathrm{real}}=20.17425+52.10569X_{0}+60.21470X_{1}+228.37659X_{5}\\ -117.3365X_{0}X_{1}-1150.8167X_{1}X_{5}\;\\ (0.01\leq X_{0}\leq 0.2,\;0.01\leq X_{1}\leq 0.2,\;0.001\leq X_{5}\leq 0.02) \end{array}$$
(26)$$\begin{array}{l} {\mathrm{ROE}}_{\mathrm{cod}}=35.29+4.92A+4.34B+0.475C-2.47\mathrm{AB}-1.59\mathrm{BC}\\ (A,B,C\in \left[-1,1\right]) \end{array}$$

Therein, Equation (25) is in terms of the actual factors and can be utilized to predict the AOR. Equation (26) is in terms of the coded factors, which is necessary for the subsequent GA. In Equation (26), the term A corresponds to *X*_0_, B corresponds to *X*_1_, and C corresponds to *X*_5_. The upper (−1) and lower limits (1) of the value range of term A correlated with the upper (0.01) and lower limits (0.2) of factor *X*_0_, which is also correspondingly applied for the terms B and C.

When the predictive equation of AOR is acquired based on the BBD design, the corresponding optimal parameters of rolling friction coefficient and surface energy can be achieved by making the predicted AOR equal to the experimental value (34.23° for the PEEK/CF_30wt%). Thus, the objective equation for the GA is expressed as
(27)$${\left({\mathrm{ROE}}_{\mathrm{cod}}-34.23\right)}^{2}=0$$

Equation (27) is input into the MATLAB GA toolbox with the value range of [−1, 1] for all corresponding variables. The results are given in Table 10. So far, all the essential factors have been determined for the powder dynamics model.

### 4.3. Experimental Verification

The calibrated DEM model for the PEEK/CF composite powders should be experimentally verified. As a comprehensive index of powder properties, AOR can be utilized to conduct the experimental verification work. The AOR experiment is carried out based on the standard GB11986-89. The experimental AOR results of PEEK, PEEK/CF_30wt%, and PEEK/CF_50wt% are shown in Figure 10a, with the detailed data listed in Table 11. The simulation results are shown in Figure 10b, and the detailed data are given in Table 12. It appears that the experimental and simulation results show a maximal deviation of 4.89%, which can indicate the high reliability and effectiveness of the developed powder dynamics model.

It also reveals that the pure PEEK possesses the largest AOR, which implies the minimum of powder fluidity. With the addition of CF, the AOR of the composite powder decreases and the powder fluidity rises. Importantly, the AOR decreases with the increase in the weight fraction of CF, exhibiting improved powder fluidity. Since the powder fluidity is greatly affected by the cohesive effect between particles of PEEK-PEEK, PEEK-CF, and CF-CF. The influence of PEEK-PEEK, PEEK-CF, and CF-CF interfacial reactions on the composite AOR is in descending order (discussed in Section 4.2). The PEEK-related cohesion plays a dominant role in the PEEK/CF powder packing. Thus, the increased volume fraction of CF leads to a decrease in total cohesive interaction between particles, resulting in improved powder fluidity.

### 4.4. Evaluation of the Powder Deposition

The investigation of the powder deposition process is necessary to achieve the expected behaviors of the laser-sintered parts in PEEK/CF composites. The surface properties and recoating difficulty of powder layers can be evaluated in a qualitative fashion using the powder deposition process. The fluctuation of the powder surface is observed by an optical microscope (VHX-6000, KEYENCE (CHINA) CO., LTD., Shanghai, China). Figure 11 shows the micro-pictures of the surface topography and undulation of PEEK, PEEK/CF_30wt%, and PEEK/CF_50wt%. It appears that the image quality of the surface morphology of PEEK is worse than the other two composite powders. This indicates a violent surface fluctuation of the PEEK powders, which is caused by the inhomogeneous particle distribution-induced porosity. With the addition of the CF, the surface quality is improved and also meliorates with the increase in the weight fraction of CF, which is caused by the particle distribution of the adopted CF. As shown in Figure 9b, numerous CF particles exhibit the size range of 5–20 μm. These small and uneven CF particles exactly fill the inter-particle cavity among PEEK powders, and the surface of its powder bed is observed smoothly.

Figure 12 shows the cross-sections of the simulated powder deposition results. It also shows that drastic undulation occurs on the PEEK surface. The surface fluctuation of PEEK/CF_30wt% is slightly mitigated and improved a lot for the PEEK/CF_50wt%, which is consistent with the experimental results. The standard deviation of the surface contour curve of the powder deposition layer is used to quantitatively characterize the fluctuation of the layer surface, which is similar to the RMS roughness. The fluctuation of the layer surface decreases with the decrease in the standard deviation, and the corresponding powder layer becomes smoother.

The experimental and simulation results of the standard deviation of the surface contour curve are shown in Figure 13 and Table 13. According to the experimental results, the surface smoothness of PEEK/CF_30wt% is increased by 31.3% compared to the PEEK powder, while the surface smoothness of PEEK/CF_50wt% powder is improved by 55.2% compared to PEEK powder. According to the simulation results, compared to the PEEK powder, the surface smoothness of the PEEK/CF_30wt% is increased by 27.4%, and the PEEK/CF_50wt% powder is improved by 50.2%. The similar experimental and simulation results further verify the reliability of the DEM model.

In summary, the method of systematic material evaluation possesses universal applicability for different types of composite powders, even multi-component composites. The powder dynamic modeling for composite powders has been developed to accurately simulate the composite powder deposition. The deposition parameters can be optimized using the DEM approach effectively.

## 5. Conclusions

This work proposes a systematic framework for material evaluation to be incorporated with the powder dynamics model for the numerical investigation of composite powder deposition. PEEK/CF as typical feedstock materials of laser sintering are investigated, and the interaction between short CF and PEEK powders is quantified upon deposition. The calibrated particle dynamic model can explain the powder flow behaviors and interactions, and it is validated by the AOR experiments with a maximal deviation of 4.89%. It shows that the AOR decreases with the increase in the weight fraction of CF. This indicates the improved powder fluidity with the addition of a proper percentage of CF with the length of 5–20 µm. In addition, the powder deposition process is investigated through the experiment and the DEM simulation. It is found that the surface quality is improved with the increase in the weight fraction of CF. The results show an experimental improvement of 31.3% and 55.2% for PEEK/CF_30wt% and PEEK/CF_50wt%, with a simulated improvement of 27.4% and 50.2% for corresponding composite powders. The similar results between experimental and simulated improvement further verify the reliability of the DEM model. Importantly, this method is generally applicable to investigate the dynamic behaviors of different types of composite powders including multi-component particles. The developed deposition model can be helpful for optimizing the deposition parameters and material composition so as to improve the packing quality of composite powders, which significantly influences the LPBF process of composite materials in additive manufacturing.

## Figures and Tables

**Figure 1 polymers-15-02863-f001:**
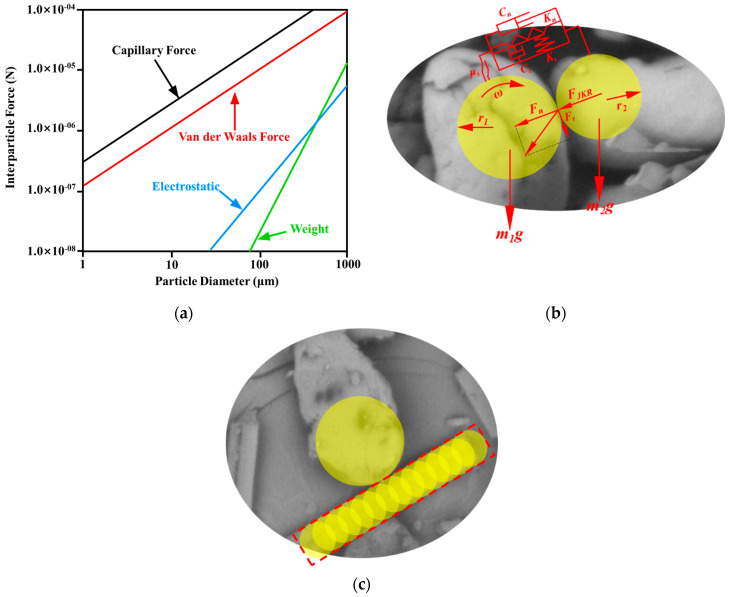
Interaction between particles and particle modeling method: (**a**) the relationship between particle force and particle diameter [30]; (**b**) contact force diagram between particles; (**c**) multiple-sphere method to model the CF particle.

**Figure 2 polymers-15-02863-f002:**
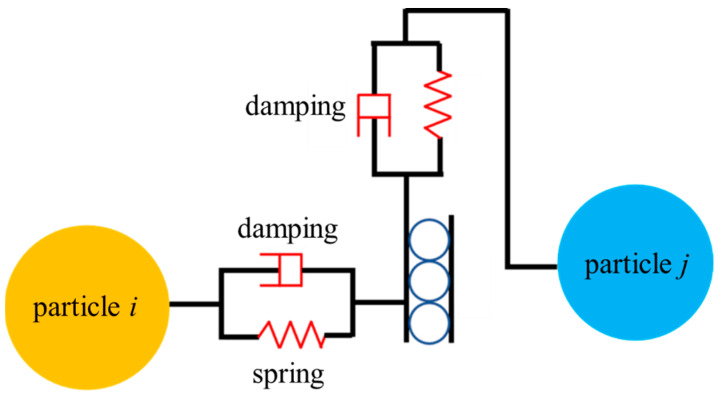
Hertz–Mindlin contact model.

**Figure 3 polymers-15-02863-f003:**
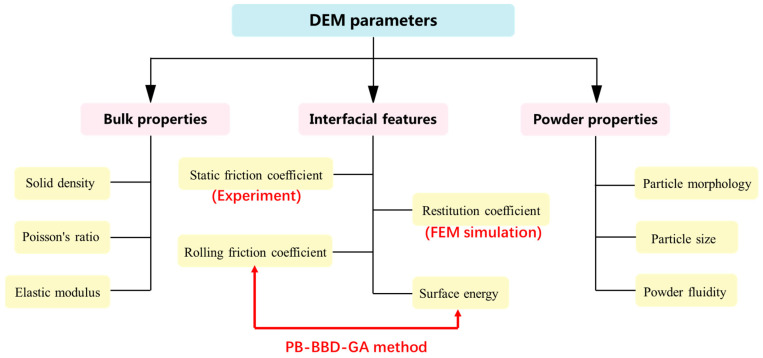
Parameter classification and calibration methods of the DEM model.

**Figure 4 polymers-15-02863-f004:**
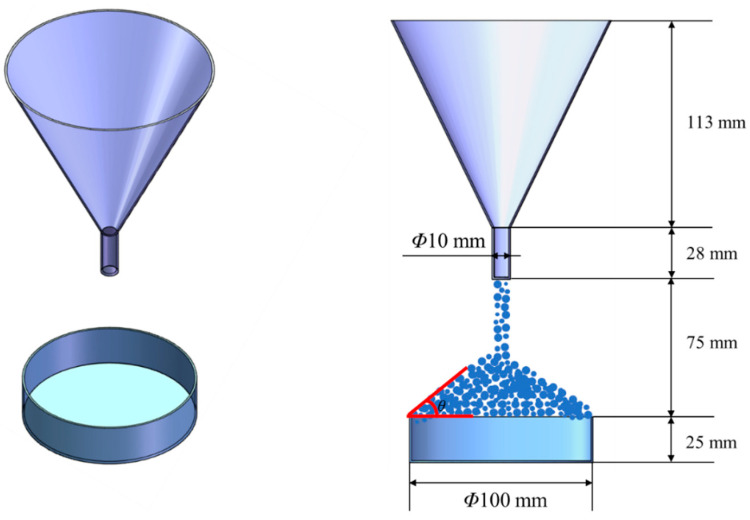
The AOR experiment according to the GB11986-89.

**Figure 5 polymers-15-02863-f005:**
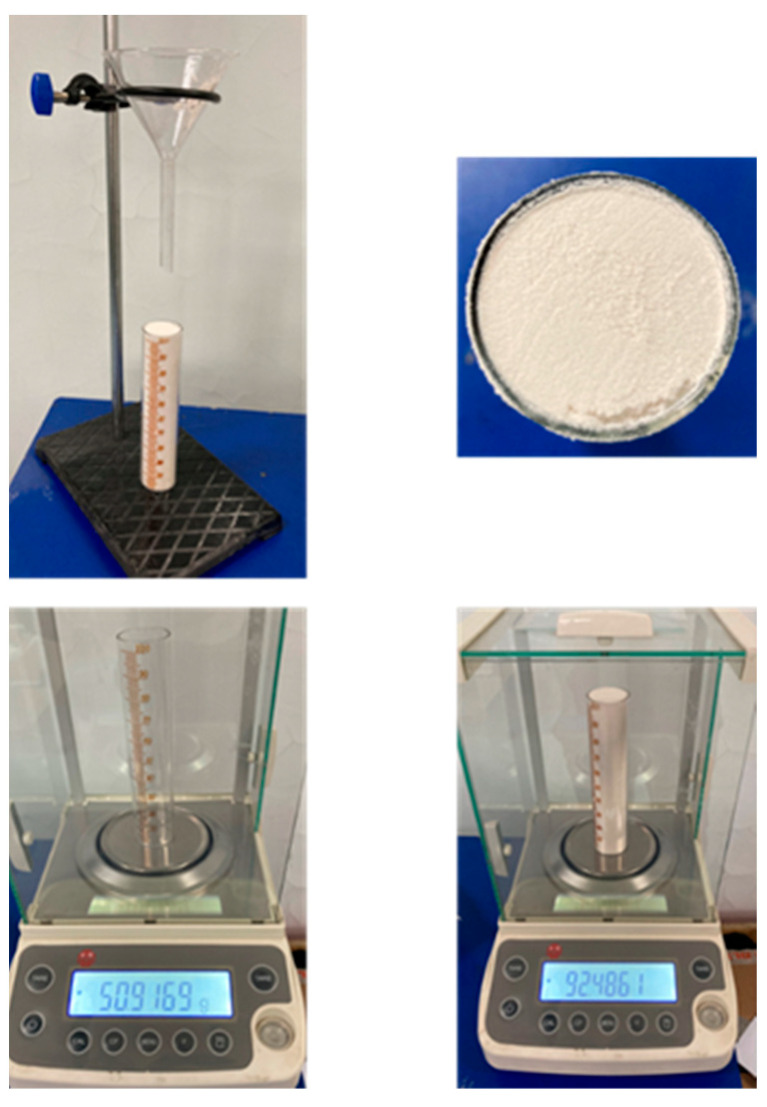
Measurement for the bulk density.

**Figure 6 polymers-15-02863-f006:**
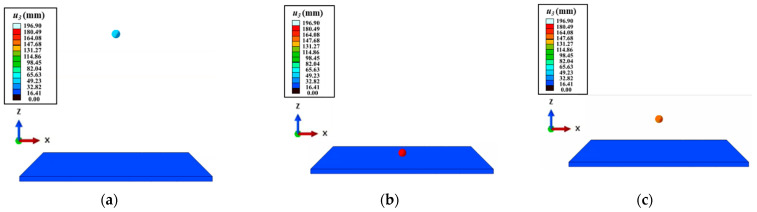
FEM simulation of the restitution coefficient: (**a**) initial moment; (**b**) contact with bottom plate; (**c**) bouncing to maximum height.

**Figure 7 polymers-15-02863-f007:**
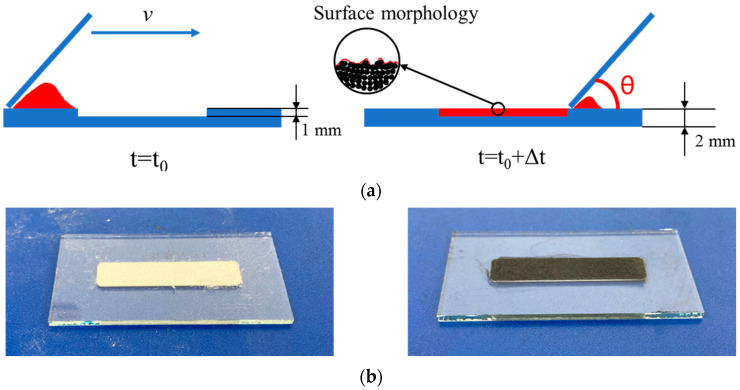
The powder deposition process of experiment and simulation: (**a**) the powder deposition process and the surface morphology observation; (**b**) the flat surface of the spread powder with the naked eye; (**c**) the observation zone; (**d**) the cross-section position of the simulated powder layer.

**Figure 8 polymers-15-02863-f008:**
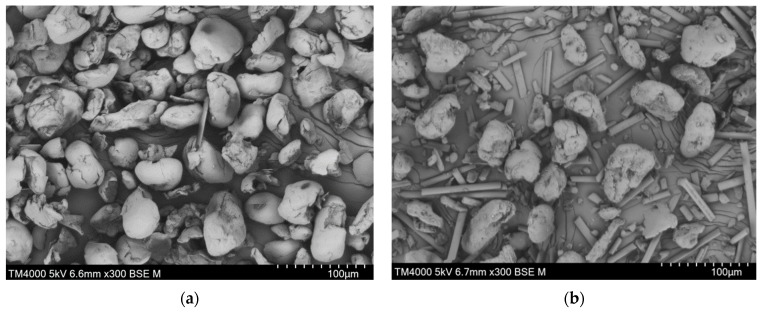
The results of SEM experiments: (**a**) particle morphology of PEEK; (**b**) particle morphology of PEEK/CF_30wt%.

**Figure 9 polymers-15-02863-f009:**
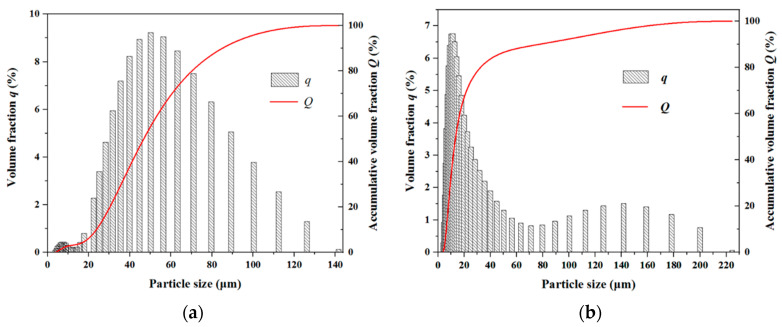
Particle size distribution: (**a**) PEEK; (**b**) CF.

**Figure 10 polymers-15-02863-f010:**
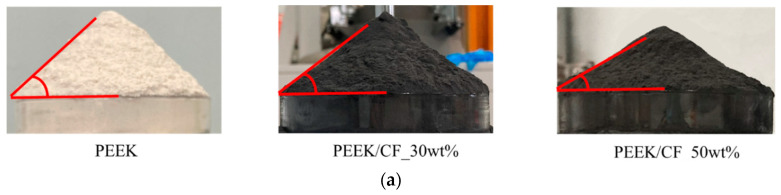
The experimental and simulation results of AORs for PEEK, PEEK/CF_30wt%, and PEEK/CF_50wt%: (**a**) experimental results; (**b**) simulation results.

**Figure 11 polymers-15-02863-f011:**
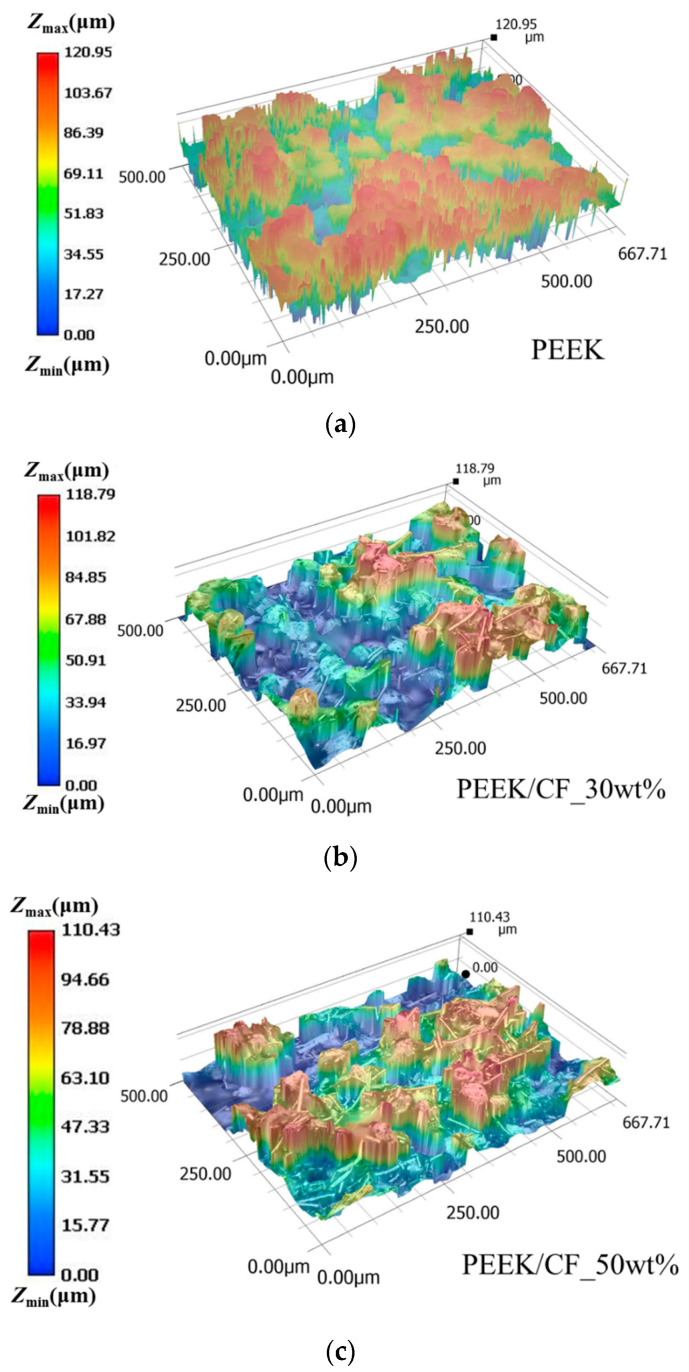
The experimental micro-pictures of the surface topography and undulation: (**a**) PEEK; (**b**) PEEK/CF_30wt%; (**c**) PEEK/CF_50wt%.

**Figure 12 polymers-15-02863-f012:**
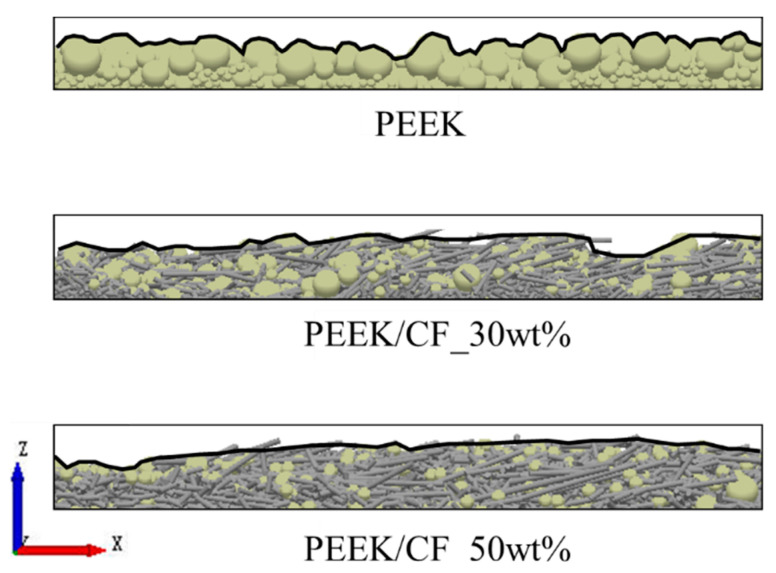
The cross-section of the simulated powder deposition of PEEK, PEEK/CF_30wt%, and PEEK/CF_50wt%.

**Figure 13 polymers-15-02863-f013:**
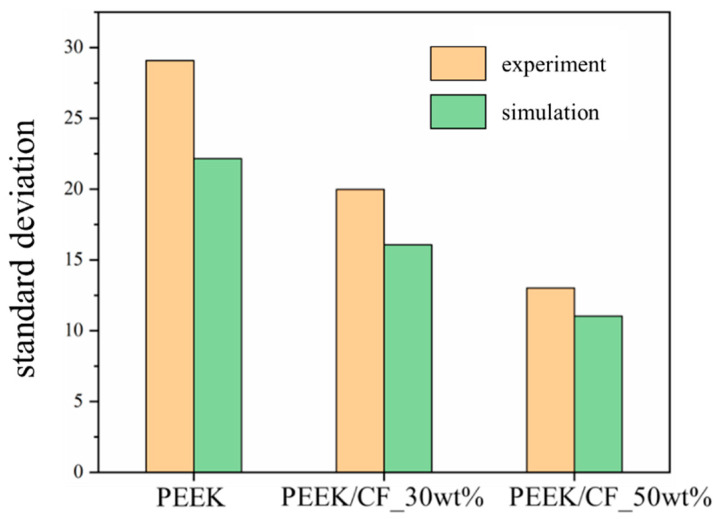
Experimental and simulation results of standard deviation of powder surface profile.

**Table 1 polymers-15-02863-t001:** The inherent mechanical properties of PEEK, CF, and glass.

Material	Elastic Modulus (Gpa)	Poisson Ratio	Density (kg/m^3^)	Reference
PEEK	3.6	0.38	1300	[42]
CF ^1^	15	0.2	1760	[43]
Glass	64	0.2	2230	ISO 3585-1998

^1^ Elastic modulus of CF is 15 Gpa along the horizontal direction.

**Table 2 polymers-15-02863-t002:** The parameters of the PB test.

Parameters	Symbol	Initial Parameter Space
PEEK-PEEK rolling friction	*X* _0_	0.01–0.2
PEEK-CF rolling friction	*X* _1_	
CF-CF rolling friction	*X* _2_	
PEEK-glass rolling friction	*X* _3_	
CF-glass rolling friction	*X* _4_	
PEEK-PEEK surface energy	*X* _5_	0.001–0.02 (J/m^2^)
PEEK-CF surface energy	*X* _6_	
CF-CF surface energy	*X* _7_	

**Table 3 polymers-15-02863-t003:** Average particle size distribution range of PEEK and CF.

Material	Particle Size (μm)	
	PEEK	CF
D_10_	23.884	6.39
D_50_	46.702	13.433
D_90_	84.546	77.043

**Table 4 polymers-15-02863-t004:** The results of the restitution coefficient and the static friction coefficient.

Type	Static Friction Coefficient				Restitution Coefficient		
	Tilt Angle α1 (°)	Tilt Angle α2 (°)	Tilt Angle α3 (°)	*μ*	*v_app_*^1^ (m/s)	*v_off_* (m/s)	*e*
CF-PEEK	25.34	24.92	25.62	0.4726	1.98	0.5336	0.2695
CF-CF	21.18	20.41	21.33	0.3833	1.98	0.1665	0.0841
PEEK-PEEK	25.13	25.42	24.82	0.4689	1.98	0.5437	0.2746
CF-glass	5.89	5.91	5.70	0.1021	1.98	0.1677	0.0847
PEEK-glass	8.20	8.34	7.85	0.1429	1.98	0.5405	0.2730

^1^ Elastic modulus of CF is 15 Gpa along the horizontal direction.

**Table 5 polymers-15-02863-t005:** The bulk density results of PEEK and PEEK/CF.

Powder	Empty Graduated Cylinder (g)	Full Graduated Cylinder (g)	Volume (ml)	Bulk Density (g·cm^−3^)
PEEK	50.9169	92.4863	100	0.4157
PEEK/CF_30wt%	51.5233	102.9567	100	0.5143
PEEK/CF_50wt%	51.6115	107.061	100	0.5545

**Table 6 polymers-15-02863-t006:** The design and results of the PB test.

No.	Rolling Friction Coefficient					Surface Energy			Virtual Parameters			AOR (°)
	*X* _0_	*X* _1_	*X* _2_	*X* _3_	*X* _4_	*X* _5_	*X* _6_	*X* _7_	*A*	*B*	*C*	
1	0.20	0.20	0.01	0.20	0.20	0.02	0.001	0.001	−1	1	−1	48.92
2	0.01	0.20	0.20	0.01	0.20	0.02	0.02	0.001	−1	−1	1	45.30
3	0.20	0.01	0.20	0.20	0.01	0.02	0.02	0.02	−1	−1	−1	46.90
4	0.01	0.20	0.01	0.20	0.20	0.001	0.02	0.02	1	−1	−1	38.62
5	0.01	0.01	0.20	0.01	0.20	0.02	0.001	0.02	1	1	−1	33.74
6	0.01	0.01	0.01	0.20	0.01	0.02	0.02	0.001	1	1	1	33.75
7	0.20	0.01	0.01	0.01	0.20	0.001	0.02	0.02	−1	1	1	42.17
8	0.20	0.20	0.01	0.01	0.01	0.02	0.001	0.02	1	−1	1	48.48
9	0.20	0.20	0.20	0.01	0.01	0.001	0.02	0.001	1	1	−1	45.81
10	0.01	0.20	0.20	0.20	0.01	0.001	0.001	0.02	−1	1	1	38.23
11	0.20	0.01	0.20	0.20	0.20	0.001	0.001	0.001	1	−1	1	42.54
12	0.01	0.01	0.01	0.01	0.01	0.001	0.001	0.001	−1	−1	−1	26.10

**Table 7 polymers-15-02863-t007:** Significance analysis of the model and corresponding parameters of the PB test.

Source	Sum of Squares	df	Mean Square	F-Value	*p*-Value	
Model	526.34	8	65.79	15.01	0.0239	significant
*X* _0_	290.94	1	290.94	66.36	0.0039	significant
*X* _1_	134.44	1	134.44	30.66	0.0116	significant
*X* _2_	17.43	1	17.43	3.98	0.1401	-
*X* _3_	4.53	1	4.53	1.03	0.3843	-
*X* _4_	12.04	1	12.04	2.75	0.1961	-
*X* _5_	46.55	1	46.55	10.62	0.0472	significant
*X* _6_	17.67	1	17.67	4.03	0.1383	-
*X* _7_	2.74	1	2.74	0.6253	0.4868	-
R^2^ = 0.9756, Adjusted R^2^ = 0.9106						

**Table 8 polymers-15-02863-t008:** The design and results of the BBD test.

No.	*X* _0_	*X* _1_	*X* _5_	AOR (°)
1	0.010	0.010	0.0105	24.21
2	0.200	0.010	0.0105	37.33
3	0.010	0.200	0.0105	37.56
4	0.200	0.200	0.0105	40.82
5	0.010	0.155	0.0010	30.29
6	0.200	0.155	0.0010	41.53
7	0.010	0.155	0.020	31.11
8	0.200	0.155	0.020	42.83
9	0.105	0.010	0.001	28.40
10	0.105	0.200	0.001	40.52
11	0.105	0.010	0.020	32.41
12	0.105	0.200	0.020	38.19
13	0.105	0.105	0.0105	34.95

**Table 9 polymers-15-02863-t009:** Significance analysis of the model and corresponding parameters of BBD test.

Source	Sum of Squares	df	Mean Square	F-Value	*p*-Value	
Model	380.53	5	76.11	66.39	7.68 × 10^−8^	significant
*X* _0_	193.51	1	193.51	168.81	5.12 × 10^−8^	significant
*X* _1_	150.82	1	150.82	131.57	1.85 × 10^−7^	significant
*X* _5_	1.81	1	1.81	1.57	0.235541	-
*X* _0_ *X* _1_	24.34	1	24.34	21.24	0.000755	significant
*X* _1_ *X* _5_	10.05	1	10.05	8.77	0.012946	significant
R^2^ = 0.9679, Adjusted R^2^ = 0.9533						

**Table 10 polymers-15-02863-t010:** The optimal parameters obtained by the GA.

Variable		Coded Value			Real Value
Coded	Real	Max	Min	Optimal	
A	*X* _0_	1	−1	−0.169	0.1305
B	*X* _1_	1	−1	0.008	0.1562
C	*X* _5_	1	−1	−0.581	0.0050

**Table 11 polymers-15-02863-t011:** Experimental AORs of PEEK, PEEK/CF_30wt%, and PEEK/CF_50wt%.

Powder	Repose Angle *θ* (°)			
	Group 1	Group 2	Group 3	Average
PEEK	41.59	41.43	41.65	41.56
PEEK/CF_30wt% ^1^	33.66	34.86	34.17	34.23
PEEK/CF_50wt%	28.29	27.93	27.37	27.86

^1^ The AOR of PEEK/CF_30wt% has been referenced at the DOE process; thus, the AORs of PEEK/CF_50wt% and PEEK are the verification targets.

**Table 12 polymers-15-02863-t012:** The comparison between simulation and experimental results of AOR.

Material	Simulation Results of AOR (°)					Experimental AOR (°)	Error (%)
	*x+* *	*x−* *	*y+* *	*y−* *	Average		
PEEK	44.54	43.70	41.98	43.11	43.33	41.56	4.26
PEEK/CF_30wt%	36.29	35.01	36.48	35.84	35.90	34.23	4.89
PEEK/CF_50wt%	25.46	29.37	28.36	28.81	28.00	27.86	0.50

* *x*+, *x*−, *y*+, and *y*− represent the directions when reading the AOR results.

**Table 13 polymers-15-02863-t013:** Experimental and simulation results of the improvement of PEEK/CF_30wt% and PEEK/CF_50wt% compared to PEEK.

Powder	Standard Deviation of the Surface Contour Curve			
	Experiment	Improvement Compared to PEEK	Simulation	Improvement Compared to PEEK
PEEK	29.07	-	22.15	-
PEEK/CF_30wt%	19.98	31.3% ↑	16.07	27.4% ↑
PEEK/CF_50wt%	13.01	55.2% ↑	11.02	50.2% ↑

## Data Availability

Data openly available. The data information involved in this work has been presented in the paper and can be publicly referenced.

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
