# Peer review of "Material Evaluation and Dynamic Powder Deposition Modeling of PEEK/CF Composite for Laser Powder Bed Fusion Process"

_polymers, 2023, doi:10.3390/polym15132863_

Round 1
Reviewer 1 Report
we received paper from polymers with the title "Material evaluation and dynamic powder deposition modeling of PEEK/CF composite for powder-bed fusion process". the paper was breakthrough process of new manufacturing process how to combine fibers with the additive manufacturing with the type of LPBF. several items need to be clarified before it goes tobe publish.
1. In the abstract, stated the best results in value mode. i.e. XX% or XXMPa, etc.
2. CF should be clearly identified with continuous or chop fibers.
3. The references of LPBF should be added related to additive manufacturing i.e. First-rate manufacturing process of primary air fan (PAF) coal power plant in indonesia using laser powder bed fusion (LPBF) technology.
4. In Fig. 1, if the real image (SEM Images) are important, it should be added with scale.
5. In materials section, the brand and the detailed type of the peek, carbon fibers, and other materials that used in the study should be added.
6. The type of the equipment that used in the study also need to be added.
7. In Fig. 10, what is the purpose to do this test?
8. What is the type of the equipment that used in fig. 11?
9. In fig. 13, why the difference between experimental and simulation is not near and seem link have gap?
10. In conclusion, it should be put in single paragraph, add all the most important results of every tests have done.
the english id not have major flaws.
Author Response
We appreciate your positive and constructive comments and suggestions very much. The manuscript has been revised accordingly. The issues addressed in the manuscript are outlined as follows. The modified areas in the manuscript is displayed in a shaded background.
Detailed information can be found in the Word/PDF document below.

Reviewer 2 Report
1. I have some doubts about the references used to obtain the mechanical properties of PEEK, and I'm not entirely sure if the elastic modulus used corresponds to the bulk material. Please revise.
2. Bulk density determination must include compacity factors to correct the apparent density from the real density.
3. I have doubts about the static friction coefficient measure; this method is valid for solid samples with low-roughness surfaces. If you are measuring static friction coefficients for powders samples, I think this method is no longer valid. Please revise.
4. Just as commentary, the standard deviation of the surface contour curve is known as RMS roughness, and with the data from surface profilometry, I think it is preferable to determine the roughness of the surface instead of just one contour profile.
There are some small grammatical or redaction mistakes that could be easily fixed (et al. should be in italics, or verb conjugation times, for example). Fixing these details will improve your manuscript's readability considerably. I strongly recommend that an English native speaker could revise this aspect.
Author Response

(The authors gave the same response as above.)

Round 2
Reviewer 1 Report
After carefully checking the revision, the authors did not revise the manuscript based on the reviewer's suggestion. For example, the first paragraph that claimed the advantages of LPBF and its applications with only 1 reference is insufficient. The previous revision already suggests adding more references, but the authors did not follow suggestions from the reviewers. Please carefully check the suggestion from the reviewer and revised accordingly.
Minor or sufficient English quality.
Author Response
We sincerely apologize to you.
Previously, we did not make significant modifications to the references as requested due to our careless.
In this version, references related to the background of additive manufacturing and LPBF have been added at beginning and the format of the references has been proofread. We hope to get your understanding and satisfaction.
Thank you so much and wish you all the best
(page:1; line:32-38)

Reviewer 2 Report
You resolve of all my doubts
Author Response
Thank you very much for your recognition of our work.
Wish you all the best!
Round 3
Reviewer 1 Report
The present version can be accepted after carefully check